# An Overview on MADS Box Members in Plants: A Meta-Review

**DOI:** 10.3390/ijms25158233

**Published:** 2024-07-28

**Authors:** Prakash Babu Adhikari, Ryushiro Dora Kasahara

**Affiliations:** Biotechnology and Bioscience Research Center, Nagoya University, Nagoya 464-8601, Japan

**Keywords:** MADS box, biotic and abiotic factors, meta-review, pleiotropic, flowering

## Abstract

Most of the studied MADS box members are linked to flowering and fruit traits. However, higher volumes of studies on type II of the two types so far suggest that the florigenic effect of the gene members could just be the tip of the iceberg. In the current study, we used a systematic approach to obtain a general overview of the MADS box members’ cross-trait and multifactor associations, and their pleiotropic potentials, based on a manually curated local reference database. While doing so, we screened for the co-occurrence of terms of interest within the title or abstract of each reference, with a threshold of three hits. The analysis results showed that our approach can retrieve multi-faceted information on the subject of study (MADS box gene members in the current case), which could otherwise have been skewed depending on the authors’ expertise and/or volume of the literature reference base. Overall, our study discusses the roles of MADS box members in association with plant organs and trait-linked factors among plant species. Our assessment showed that plants with most of the MADS box member studies included tomato, apple, and rice after Arabidopsis. Furthermore, based on the degree of their multi-trait associations, *FLC*, *SVP*, and *SOC1* are suggested to have relatively higher pleiotropic potential among others in plant growth, development, and flowering processes. The approach devised in this study is expected to be applicable for a basic understanding of any study subject of interest, regardless of the depth of prior knowledge.

## 1. Introduction

The current study is not a typical review in a canonical sense, in which the content of the article would largely depend on the expertise of the author. The contents of this article primarily rely on data derived from the MADS box-related manually curated local reference database, hence the term ‘meta-review’ in the article title. This study has attempted to establish a direct, literature-based approach to conducting a literature review using relevant search keywords, and constraint terms. While this study took MADS box studies as a test case, the devised approach is expected to be applicable to any other study-to-(key-of-interest) associations.

MADS represents the four fabulous founder homeotic proteins, MCM1 (from *Saccharomyces cerevisiae*), AGAMOUS (from *Arabidopsis thaliana*), DEFICIENS (from *Antirrhinum majus*), and SRF (from *Homo sapiens*), which was first observed and annotated as such by Schwarz-Sommer, et al. [1] based on the high similarity of the domain among the proteins. Studies suggest that a common MADS box ancestor of fern and seed plants already constituted at least two flowering-related MADS box genes (MIKC-type), approximately 400 MYA [2,3]. Several prominent studies have been conducted on the phylogenetic classification of MADS box gene members [4,5,6,7,8,9,10,11]. These studies suggest the diverse roles of genes during plant growth and development. While the majority of studies are associated with flowering, a comprehensive understanding of the genes would offer a broader perspective on their functional evolution and diversification. A large number of independent studies on MADS box member genes are available in models as well as non-model plants. The utilization of their holistic information in a single manuscript is relatively daunting, yet seems essential to have an ‘aerial’ perspective regarding the progress on the subject, which may offer initial ground for experimental design to the experts and non-experts alike. With this intent in mind, we carried out a meta-review of MADS-associated studies.

Here, we have discussed the MADS box member genes and their roles in plant growth phases, the interplay of known members with hormonal cues, and the potential involvement of known members in bridging multiple traits and/or factors based on the information retrieved from the curated reference database containing 773 independent studies.

## 2. Study-Based Meta-Review: Basic Strategy

Since their discovery in 1990, there have been many studies on MADS box genes in several plants, as shown in Figure 1 [12]. Most of those studies have been meticulously planned, conducted, peer-reviewed, and published. Using them as a direct reference to gain a broader understanding of the roles of genes-of-interest in plants would offer an advantage to researchers in study design, regardless of their depth of knowledge in the study subject at the beginning. With such a concept in mind, references were fetched from PubMed, Google Scholar, and Semantic Scholars using main keyword ‘MADS box’ with or without any of the additional keywords—‘flowering’, ‘genome-wide‘, ‘vegetative‘, ‘seed germination‘, and ‘seed development’. They were screened for gene-specific experiments, excluding most of the broader studies, such as genome-wide studies and reviews, except for tissue- and/or gene/clade-specific studies. The names of the studied organisms were manually extracted from the remaining 773 references (published from 1992 to 2024) (Appendix A) and used for the gene-to-study association analyses. We used an in-house script to pool keywords/search terms with or without constraints and generated a word cloud for each gene pool using the wordcloud 1.9.3 python library [13]. For general gene IDs, we used the Arabidopsis MADS box member gene IDs and their respective synonymous IDs as search keywords. For cross-species-specific MADS box gene screening, we used wildcards. The complete list of gene ID search keywords, wildcards, and organism and plant organ search terms is provided in Appendix A. The constraints used while screening the literature on a specific gene pool have been provided in Appendix A. A threshold of 3 was set during the analysis to reduce potential false-positive hits, unless mentioned otherwise. While visualizing the data, the IDs with hit numbers below the threshold were grayed out to make the IDs with above-threshold hits stand out.

## 3. General Overview: Plants and Genes of Studies

Our very initial question to the database was, ‘in which plant the MADS box member genes have been studied the most?’ Being a model plant, it was not unusual for Arabidopsis to appear as the first. In addition to additional model plant Nicotiana, cereals (rice, maize, wheat), vegetables (tomato in particular), fruits (apple, peach), and ornamental plants (orchids) were some of the top hits. Rice, tomatoes, and apples have frequently been used as model plants in monocot, vegetable, and fruit tree studies. In total, 188 organisms were recorded as having been studied directly or indirectly by the studies used for the analysis (Figure 1a). Since Arabidopsis is the most studied plant, we next examined the most studied MADS gene members amongst the studies. Interestingly, most studied genes were among the known flowering repressors (*FLC*, *AGL15*, *AGL24*, *SVP*, *AGL18*, *MAF3/4/5*) and promoters (*STK*, *SEP3*, *AG*, *AGL17*, *PI*) (Figure 1b). Since several gene MADS box IDs were identical to their associated clade IDs, such genes often showed higher hits (Figure 1b and Appendix A). Additionally, when checked using only clade IDs, SVP returned the highest hits, followed by AP1, SOC1, FLC, AG, SEP, and AP3 (Appendix A). Interestingly, our analysis showed a stark disparity between the studies on type I and type II MADS box members. Among the 69 type I members, only 15 showed study association hits, while 41 showed such hits among only 46 type II members (Appendix A). We further checked the potential MADS box members studied in other organisms using wildcard keywords for MADS, AGL, DAM, RIN, and other related genes. Among the 560 putative gene terms retrieved, *StMADS11*, an *SVP* member from potato, and *VRN1*, a *FUL* homolog from wheat and relatives, were the most studied genes followed by *TAGL1* (an AG clade member from tomato), *OsMADS1* (a SEP clade member from rice), and *TM6* (an AP3 clade member often from tomato) (Figure 1c).

## 4. Screening Potential Pleiotropics: Sorting Threads from Haystack

The majority of genes closer to the terminal end of genetic/physiological pathways often tend to be less pleiotropic in nature due to their narrowly specified functions and, hence, generally tend to be associated with a single if not closely associated trait/s. In contrast, pleiotropic genes often have multi-trait associations. To retrieve such MADS box members, we initially set out to assess the tissue-to-study association for roots, shoots, leaves, apical meristem/SAM, flowers, fruits, and seeds. Subsequently, we checked for recurring genes among the independently sorted gene pools (Figure 2), which are presumed to have pleiotropic functions in plants.

Most of the highest-hit IDs were of such mutual IDs (e.g., *FLC*, *SOC1*, *SVP*, *AP1*, *AG*, *AGL15*), as observed in earlier cases, except for *FUL*, *AGL17*, *SEP1*, and *AGL24* in the Arabidopsis gene tree (Figure 2a). Our results align with the pleiotropic role of genes during plant growth and development. Taking *FUL* as an example, initial observations made on the mutants of the FUL gene reported disorder in silique (fruit) development (shorter silique with frequent premature dehiscence) in Arabidopsis due to the absence of cell expansion and selective restriction of cell division. The mutant siliques at the mature stage contained highly compacted seeds within the short silique—hence the name ‘*FRUITFULL*’ [14]. The study also reported a difference in cauline leaf shape (more round in the mutant). Latter studies have shown that *FUL* directly represses downstream MADS box members *SHP1* and *SHP2*, which is crucial for the lignification and formation of the silique dehiscence region as the siliques remain ‘shatter-proof’ in their cumulative mutants [15,16]. In addition to its role in fruit development, *FUL* has been associated with other functions. Such examples include its involvement in meristem determinacy by negatively regulating AP2 in the developing inflorescence [17], and in apical hook opening modulation by negatively regulating the expression of growth-promoting genes in Arabidopsis [18]. Additionally, the involvement of its homolog from birch [19] and several other plants in precocious flowering has been observed. Yet, additional instances include the role of the rice *FUL* homolog in normal seed development by regulating at least two key genes involved in starch synthesis, *OsAGPL2* and *WAXY* [18], and the crucial role of its tomato orthologs *FUL1* and *FUL2* in tomato fruit ripening, potentially by forming a tetramer complex with additional MADS box members RIN and AGL1 [20]. The tomato *FUL1/2*, along with an additional MADS member *MBP20* (a SEP-like gene), has also been attributed with roles in vegetative-to-reproductive transition and inflorescence architecture regulation [21]. In rice as well, *AP1/FUL* homologs (*OsMADS14*, *OsMADS15*, and *OsMADS18*) and a *SEP* homolog (*PAP2*) reportedly confer floral meristem identity [22]. Another study further reported the ABA responsiveness of *OsMADS18* and its involvement in various developmental features including germination, tillering, and inflorescence architecture [23]. In soybeans, a study reported the involvement of a *FUL* homolog haplotype, *GmFULa*, in plant biomass and seed yield without affecting flowering time [24].

Among the cross-species MADS box gene pool (Figure 2b), *STMADS11* and *TAGL1*, both tomato-derived gene IDs, had the highest hits. It should be noted that, similar to several Arabidopsis gene IDs, *STMADS11* has often been used as a clade ID (synonymous with SVP clade). Nevertheless, the gene itself has been attributed to diverse developmental roles in plants. Here, while taking *TAGL1* (an AG clade member and SHP homolog) as a test example, unlike that observed for the Arabidopsis gene ID-derived top hits, its gene-to-phenotype coverage was relatively narrow, most likely due to the lower threshold (two hits) and lax parameters used during the screening process of the cross-species-derived gene IDs. *TAGL1* has been attributed to its direct involvement in the regulation of chloroplast synthesis [25] and fruit ripening [26] in tomato; its potential involvement in tomato seed size control via interaction with another MADS box member, *SlMBP21* [27]; and its potential involvement in ethylene biosynthesis and carotenoid accumulation in ripening fruit via interaction with yet another MADS box member, SlCMB1 [28]. Because of the relatively higher hits accompanied by stronger reliability for Arabidopsis-derived gene IDs as compared to the cross-species-derived IDs, we carried out downstream analyses using the former.

## 5. Gene-to-Major Tissue Growth Associations

### 5.1. Shoots

There are not many shoot-focused studies on MADS box gene members. Our analysis with shoot/stem keywords and some exclusion terms (shoot meristem, shoot apex, and stem cell) returned 30 MADS box members with direct/indirect study associations with shoots, of which only 7 were above the threshold (Figure 3a). We compared our gene pool with the tissue-specific expression analysis derived from the pools of Parenicova, et al. [29]. Even though the study showed several type I MADS box genes expressed in the shoot, our analysis returned none. This was mainly because of the fewer studies on type I MADS box members (Appendix A), which skewed the local reference database towards type II members. In addition, our analysis returned only 14 of 23 shoot-expressed type II members, as reported by Parenicova, et al. [29]. Such disparity was expected because the analytical approach and objectives were different for these studies. Interestingly, of the seven genes above the threshold, only two (*FUL* and *AGL24*) were common in the reported study. The former reportedly affects the branch angle by negatively modulating the expression of *SAUR10* and influences other genes involved in hormone and light signaling pathways in Arabidopsis [30]. *AGL24*, on the other hand, is a flowering promoter, and its overexpression lines flower at much shorter heights, as in the case of the majority of other MADS box members promoting precocious flowering. A previous study showed that the phenotype of the *svp* mutant is epistatic to *agl24*, as the genes are involved in recruiting the co-repressor complex [31]. *SVP* and *FLC* are the key MADS box members associated with the positive regulation of vegetative growth in plants, which is often positively correlated with shoot growth. Regarding *SOC1*, a positive role of its ortholog, *MADS12*, in accelerated shoot growth was reported in poplar by its repressive effect on *GA2ox4*, a negative regulator of shoot growth [32]. *AP1* and homologs, on the other hand, act in the opposite spectrum as reported for *AP1*/*FUL* ortholog *PgMADS1* in *Panax ginseng* [33]. *ANR1*, a MADS member involved in nitrate regulation and signaling in roots, confers a positive effect in shoot growth as reported in an Arabidopsis study [34].

### 5.2. Leaves

Fifteen out of thirty-six MADS box IDs showed above-threshold hits in the leaf-associated gene pool, suggesting their potential direct or indirect involvement in leaf or leaf-associated growth and development processes in plants. It included most of the FLC clade members, even though only *FLC* had an above-the-threshold hit. Among all, *SVP* and *AP1* were the two members with the excessively higher hits (Figure 3b). While there are no reports on the involvement of *AP1* in leaf-associated features in model plants, some cross-species studies suggest its potential if not whether the gene has been neofunctionalized in them. One such study in barley reported that *PHOTOPERIOD-H1* (*Pdp-H1*), a *PRR7* gene encoding a component of the circadian clock, regulated reduction in leaf size and the number was correlated with the *Pdp-H1*-dependent induction of barley *AP1* and *FUL-like* homologs *BM3* and *BM8*, indicating their potential involvement in the process [35]. Regarding *SVP*, a study on Arabidopsis mutants reported that its dysfunctional state leads to changes in leaf size [36] and leaf shape before the formation of the first flower in addition to altering the number of rosette and cauline leaves [37].

A common feature observed with the flower promoting genes is that the transition from vegetative-to-reproductive phases often directly correlates with higher-density trichome development at the abaxial side of the cauline leaves. A study reported that *AG*, one of the IDs with hits above the threshold, is directly involved in repressing the development of the branched trichome, a key aspect of leaf development, in the gynoecium [38]. This happens by regulating cytokinin responses and genetically interacting with *KANADI1*, an organ polarity gene, suggesting that the genetic program for leaf development has been rewired during the flower formation process mainly via MADS box member floral homeotic proteins [39]. An additional study has reported that the normal expression of *AGL15*, *AGL18*, *AGL24*, and *SVP* is essential to block floral programs in vegetative tissues. In the absence of these, leaves exhibit aberrant morphology (upward curling) due to the de-repression of *FT*, a known florigen, and a MADS box member, *SEP3* [40].

*AGL6*, a member of the gene pool, reportedly affects leaf movement—an active process that regulates its circadian clock in plants—by modulating the expression of *ZEITLUPE*, a blue-light photoreceptor that governs circadian rhythm and represses photoperiodic flowering [41]. Regarding *AGL24*, an additional member with an above-threshold hit in the gene pool, a recent study demonstrated that it promotes floral organ identity speciation via the long-distance movement of its mRNA from the leaf to shoot apex. Furthermore, its encoded protein is actively degraded within the leaf itself to prevent the misexpression of its downstream genes in the tissue [42].

### 5.3. Roots

Our analysis-derived root-associated gene pool encompassed all known root-expressed or root-specific genes [43,44], except for *SHP1* and *SHP2* (Figure 3c). However, some of them showed hits below the threshold, which included *AGL18*, *AGL26*, *AGL42*, and *AGL56*. Interestingly, our analysis returned additional MADS box members with hits above the threshold, which include *FLC*, *SVP*, *AP1*, *AGL6*, *AGL15*, *FUL*, *AG*, *SEP1*, and *AGL24*. This occurrence is supported by other studies such as those on sweet potato for *SVP* and *AGL24* [45], *Medicago sativa* for *AGL6* [46] and *AP1* [47], and Arabidopsis for *FLC* [48] and *AGL15* [49], among others.

Commonly known root-associated genes have been well described by some published reviews [44,50]. The review by Alvarez-Buylla, et al. [44] discusses the root-associated MADS box members holistically among different plant species, while the study by Kim, et al. [47] particularly focuses on MADS box members associated with sweet potato root development. Nevertheless, several of the members discussed in the studies overlap. To briefly mention known functional roles of some of the MADS box members, *ANR1* and *AGL21* are involved in nitrate foraging-dependent lateral root growth and development [51,52]; *FYF*/*AGL42*, despite its unclear functional relevance, is often used as a quiescent center marker due to its exclusive expression pattern in the tissue [53]; *AGL17* exhibits its highest expression with yet unknown function in roots [54,55]; *AGL16* reportedly confers stress tolerance during root elongation [56]; *XAL1* and *XAL2* are involved in root meristem proliferation and patterning by modulating auxin transport [57,58]; *AGL15* may play a role in ROS signaling in developing roots. Other root-associated MADS members confer a more indirect effect on root growth and development.

### 5.4. Apical Meristem

A research-based report earlier from 2002 showed that the majority of the MADS box members are expressed in the Arabidopsis shoot apical meristem (SAM) among the assessed genes [59]. Later studies have further expanded the range. However, our analysis with the SAM-to-studies association returned only 16 member genes, among which 6 were above the threshold. These include *SOC1*, *AGL24*, *AP1*, *SVP*, *FUL*, and *FLC* (Figure 4a). *SOC1* and *SVP* have been implicated in the dynamic regulation of gibberellin biosynthesis and catabolism by increasing the cell size and number at the site during the transition from the apical meristem to floral meristem in Arabidopsis [60]. Furthermore, according to an earlier study, SVP and AGL24 can redundantly dimerize with AP1 to recruit the LUG-SEU co-repressor complex, thereby repressing a class E member (*SEP3*), class B members (*PI* and *AP3*), and class C members (*AG*) during the transition process to prevent precocious floral meristem differentiation [61]. *FUL*, on the other hand, has been attributed with a role in global proliferation arrest of active meristems by directly repressing members of the *AP2* clade, the ERF members. This negatively regulates the flowering and flower development process, which would otherwise repress the repressors of *WUSCHEL*, a key gene involved in meristem maintenance [62]. Regarding *FLC*, its regulation of maintaining the vegetative state of the apical meristem is at least partly mediated through the repression of its target gene *TFS1*, a B3-type REM member gene. Furthermore, *SVP* acts redundantly with *FLC* in the process. In another case, SOC1 recruits REF6, a histone demethylase, and BRM, the SWI/SNF chromatin remodeler ATPase, to activate *TFS1* during floral transition [63].

### 5.5. Flowers

The majority of the MADS box member-associated studies are flowering-focused. Hence, the flower/flowering-associated gene pool encompassed the gene IDs with the highest hits among all assessed gene pools in this study. In total, 40 genes were returned, 31 of which were above the threshold (Figure 4b). Except for two, all members belonged to the type II group. Interestingly, regardless of the threshold, all but one member (*GOA*) have reported florigenic function. Additionally, despite their absence from the derived pool, the missing type II members (*AGL30*, *AGL33*, *AGL65*, *AGL66*, *AGL67*, *AGL79*, *AGL94*, and *AGL104*) reportedly have florigenic potential. The absence of related studies in our local database could be the reason for such an occurrence.

The functional roles of type II members in floral induction have been extensively studied, and there have been well-versed evolutionary and review studies on this topic. Some of them include Gramzow and Theissen’s work [64] on both functional and evolutionary aspects of MADS box members, and simultaneous independent studies by Becker and Theissen [65] and Nam, et al. [66] on detailed dated evolutionary studies regarding MADS box gene origin and divergence. We briefly touched on this topic in our earlier review [67]. To describe the functional roles of some of the representative MADS box members in floral development, we will simply use the ABCDE model often taken as a reference in flowering-associated studies. Sepal, petal, stamen, carpel, and ovule development depend on the A−, A + B−, B + C−, C−, and C + D− function genes, respectively, in association with an E-function member. In Arabidopsis, *AP1* functions as A; *AP3* or *PI* functions as B; *AG* functions as C; *STK*, *SHP1*, or *SHP2* functions as D; and either of the *SEP* members functions as E-class genes. Another study has proposed AGL6 members, which are closely clustered with *SEP* members (Appendix A), as additional putative E-class genes based on their functional analyses in petunia, maize, and rice [68]. As described in earlier sections, several MADS box members in the gene pool play a role in floral transition, inflorescence architecture regulation, and floral meristem modulation.

### 5.6. Ovules

Our analysis returned 35 MADS box members to have study associations with ovules, among which 8 showed above-threshold hits (Figure 4c). However, the majority of the genes with at least two hits (21 in total) reportedly have direct or indirect functions in ovule development. The genes from the lower-hit spectrum (with two hits each) included *SVP*, *FEM111*/*AGL80*, *DIA*/*AGL61*, *AGL23*, *SOC1*, *AP3*, *PI*, and *GOA*. As mentioned earlier, the SVP-AP1 dimer reportedly forms a repressor complex by recruiting the co-repressors SEU-LUG and represses the expression of one of the ovule identity genes—*STK*—in floral meristems by binding to its promoter. This process is mediated by BASIC PENTACYSTEINE (BPC) transcription factors, which potentially bring changes to the bound promoter region during the repression process [69]. A study on *Ginkgo biloba*, one of the oldest living tree species, reported that the ectopic expression of its natively flower and ovule-expressed AP1/SQUA clade member *GbMADS9* downregulates its *SVP* homolog [70], roughly indicating a potential state of a similar *SVP* repression mechanism during ovule development. In other cases, *AGL61*/*DIA* and *AGL80* are crucial MADS box members for central cell development [71,72]; *AGL23* plays a crucial role in female gametophyte development, and its dysfunction renders the ovule sterile [73]; *SOC1* reportedly binds to the promoter of the *SUPERMAN* (*SUP*) gene encoding C2H2-type zinc finger protein, which is involved in cell proliferation in the ovule, in addition to its similar role in stamen and carpel primordia [74,75]; *AP3*, even though it is a B-class member, plays a crucial role in ovule development, and defects in the gene lead to the development of ovules outside their native site of development [76].

### 5.7. Pollen

A total of 26 genes were returned in the pollen-associated gene pool, among which 4 belong to the type I MADS box group. Interestingly, however, none of the members in the pool had a hit value above the threshold (Figure 4d). Apparently, there were not many detailed studies regarding the roles of MADS box members in pollen development. Nevertheless, a study in Arabidopsis reported that AGL13, one of the member genes in the pool, plays a role in anther, pollen, and ovule development, potentially by forming heterodimers with other MADS box members such as AP3, PI, and AG since it cannot form homodimers [77]. Furthermore, the study showed that *AGL13* affects the expression of *AG*, *AP3*, and *PI* via a positive feedback loop and represses its own expression by activating its repressor, *AGL6*. An additional study on Chinese fir reported a relatively upregulated status of *AP3*, *PI*, and *AGL15*; downregulated status of *SVP*; and non-differential expression of *AG* in male cones compared to female cones [78], suggesting their functional relevance in male and female cone development. An *AGL15* ortholog, *AGL18*, has also been reported to exhibit expression in developing Arabidopsis pollen at the time of mitosis, with even stronger expression later during the maturation stage, in addition to being expressed in the developing female gametophyte and endosperm [48].

### 5.8. Seeds

The seed-associated gene pool contained 41 MADS box members in total, among which 18 returned above-threshold hits. Overall, the pool encompassed 10 type I members and 31 type II members, with only 2 of the former (*AGL62* and *PHE1*) above the threshold. Some of the genes from the lower spectrum above the threshold include *SEP2*, *SOC1*, and *FLC*. Among them, *SEP2* mainly plays a role in floral development, and as reported in a study in cotton–tobacco, there is its downregulation along with other florigenic MADS box members *AP1*, *AP3*, *AGL8*, *AGL6*, and *SEP1*, upon the ectopic expression of the seed yield-enhancing gene *GhKTI12*, an elongator-associated protein-encoding gene, in tobacco [79], suggesting a negative feedback signal from the developing seed on the expression of the genes associated with floral development. Such a case aligns with a grape-tomato study that reported a decrease in the seed size and number of tomatoes upon the ectopic expression of the grape-derived *SEP2* homolog *VvMADS39* [80]. The negative effect of seed-derived signals on inflorescence architecture and fruit/seed yield has been observed in Arabidopsis [62], field pea [81], and rapeseed [82] by modulating the expression of *FUL* and *AP1*, two of the MADS box members with higher hits in the seed-associated gene pool (Figure 5a). Regarding *SOC1*, a study reported a failure of seed development in Arabidopsis lines that constitutively expressed the gene [83]. However, SOC1 clade members have potentially neofunctionalized and subfunctionalized roles in Arabidopsis flower development [84], flower senescence [85], and seed development, as reported in *Medicago truncatula* [86], barley [87], and other species.

The seed-associated gene pool additionally contained several other members that are also associated with their positive and negative regulation on flowering. Their expression in plants was expected to have respective negative and positive correlation to the seed yield. Such a case has been observed for *FLC* homologs in barley [88] and an *SVP* homolog (*SVP-A1*) in *Triticum ispahanicum* [89]. Within the developing seed itself, a soybean genome-wide expression study observed the elevated expression of AG, SEP, and FLC clade members when assessed at the globular, heart, cotyledonary, and early maturation stages [90], suggesting their positive regulatory role in the seed development process, even though their direct role in the process has not been reported yet except for *FLC*. The expression of *FLC* peaks at seed maturity is unlike *FT*, *SOC1*, and *AP1*, which reportedly show opposite expression trends with seed maturity in Arabidopsis. The seed-expressed *FLC* confers the risk aversion of the seeds after maturity by controlling germination based on ambient temperature through the modulated expression of hormonal genes [91]. Additional notable MADS box members in the gene pool include *AGL15*, which is reportedly involved in phase transition from seed maturity to germination and seedling growth. *AGL15* repression brought upon by HSI2/VAL1, a B3-domain protein, leads to the downregulation of seed maturity-associated genes by depositing the H3K27me3 at the AGL15 locus. The study further observed interaction between HSI2 and MSI1, a PRC2 repressive complex member, and suggested potential recruitment of MSI1, by HSI2, to form a PRC2 nucleation site at the *AGL15* promoter [92].

There are several studies on MADS members associated with seeds. To name a few of them here, Bemer, et al. [93] carried out an extensive assessment of the expression patterns of type I MADS box members in ovules and developing seeds. Some of the seed-expressed MADS box members reported in the study include *PHE1*/*2* (early developing endosperm), *AGL28* (developing embryo), *AGL46* (developing endosperm), and *AGL35* (chalazal endosperm), among others. Ehlers, et al. [94] reported on the roles of *SHP1* and *SHP2* in endosperm formation and seed coat development in developing seeds. Coen, et al. [95] observed that *TT16* and *STK* act as master regulators of sub-epidermal integument cell layer patterning in developing seeds.

### 5.9. Fruits

In total, 28 MADS box members were returned in the fruit-associated gene pool, which contained all but one type II member. Thirteen of them—all type II members—were above the threshold with *SVP*, *FUL*, and *AP1* at the highest and *AGL15*, *SEP3*, and *SHP2* at the lowest spectrum above the threshold (Figure 5b). Some of the members in the gene pool reportedly have a relatively subtle and indirect effect, which include *SVP*. As reported in self-abscission apple, its *SVP* homolog *MdJOINTLESS* is associated with the abscission zone often developed in the pedicel of the lateral fruits and suggested its potential involvement in regulating the auxin gradient in the developing fruit [96]. A similar case has been attributed for its tomato homolog regarding flower and fruit abscission zone development [97,98]. Among some of the genes from the lower spectrum, *SEP3*—a gene often linked with flowering promotion—plays a dynamic role in pollination-dependent fruit growth and contributes to fruit ripening as reported in strawberry [99]. It should be noted that similar to a seed set, a fruit set and its growth exert a negative effect on floral induction [100], suggesting potential involvement of flowering-related MADS box members present in the fruit-associated gene pool in the fruit-dependent feedback loop. Regarding *AGL15*, it affects the fruit maturity process if rendered active during fruit development as observed in the transgenic Arabidopsis with its constitutive expression [101]. Those plants exhibit the retention of petals and sepals long after pollination (and silique development) and bring significant delays in fruit/silique and seed maturity/desiccation. The latter study by the group further showed that such delayed floral organ senescence is correlated with the increase in *AGL15* expression around the time of floral opening, before the onset of senescence and abscission [102]. Embryo-expressed *AGL15*, however, confers no significant effect on seed desiccation.

Some of the published studies dedicated to fruit-associated MADS box members include Busi, et al. [103] profiling MADS box members during tomato fruit and seed development, Wang, et al. [104] profiling MADS box members during longan flower and fruit development, and Li, et al. [105] reviewing the MADS box member-regulated fruit ripening process in tomato and other fruit crops. The study by Busi, et al. [103] observed the expression of *TAGL12*, an *XAL1* ortholog, at different stages of fruit development. The study by Wang, et al. [104] suggested crucial roles for *DlSTK*, *DlSEP1/2*, and *DlMADS53* (putative *AGL62* and *DIA* ortholog) in the *Dimocarpus longan* fruit growth and ripening. The study by Li, et al. [105] discusses several MADS box members that are grouped within the gene pool of the current study.

### 5.10. Seed Germination

We chose seed germination instead of seedlings to pool the MADS box members potentially involved in transitioning seeds to seedlings. Seed germination returned 18 MADS box members in total, among which 2 (*AGL35* and *PHE1*) belonged to type I. Interestingly, however, none of the members returned hits above the threshold (3) (Figure 5c), which could be because of relatively less studies on this aspect of their role. *FUL*, the only member with the threshold hit, reportedly plays a positive role in seed germination as the downregulation of its *AP1*/*FUL* homolog *OsMADS18* causes delay in germination and a lower germination rate in rice [23]. The study further showed that its overexpression lines exhibit reduced auxin content and a diminished expression of strigolactone signaling-associated genes, *D14* and *OsTB1*. The expression of *OsMADS18* was positively affected by ABA, which triggered the re-localization of otherwise plasma membrane-localized MADS18 protein to the nucleus [23]. An earlier growth architecture-focused study additionally reported that *FUL* represses the expression of *SAUR10*, an auxin- and brassinosteroid-inducible gene in Arabidopsis [30]. However, a recent rice study observed a slightly reduced germination rate in the *ossaur10* mutants even though not all transgenic lines exhibited significantly different germination rates (as compared to WT) [106], indicating potential of the *SAUR10*-independent *FUL*-regulated genetic network in seed germination.

Among other members in the gene pool, the role of *AGL15* in the germination process has been discussed earlier. *ANR1* and *AGL21* act synergistically to repress seed germination in response to ABA and salinity to avoid germination at the unfavorable condition. The process is facilitated by the respective regulation of *ABI3* and *ABI5* by *ANR1* and *AGL21* [107,108]. *FLC* affects seed germination and dormancy; however, studies on its role in the process have been contradictory [109], suggesting a potential yet unknown variable mediating the *FLC* effect. *AGL16* on the other hand hinders Arabidopsis seed germination at higher salinity but suppresses ABA sensitivity during the process [110]. The suppression of its targets *HEAT SHOCK TRANSCRIPTION FACTOR A6A* (*HSFA6A*) and *MYB102* by binding to CArG elements of their respective promoters is associated with the reduced germination under a salt stress condition and ABA treatment, respectively. It is notable that similar to OsMADS18, HSFA6A localizes to the nucleus at a stress condition, which would otherwise exhibit cytoplasmic localization [111].

*TT16*, a MADS box member involved in the pigmentation of the seed coat, contributes to seed dormancy by maintaining a normal seed coat. When it is defective, the seeds exhibit premature germination in Arabidopsis [112]. A papaya study additionally showed that its *TT16* ortholog and *FUL*/*AGL8* ortholog exhibit higher expression during germination, suggesting their potential roles in the process [113]. Additional MADS box members, *STK* and *GOA*, in combination with an auxin response factor, *ARF2*, control polyamine accumulation and mucilage release in the seed coat. *STK* in particular controls pectin methylesterase (PME) activity and pectin maturation, a defect in which leads to delay in germination at a drought condition in Arabidopsis [114]. *STK* apparently contributes to salt and oxidative stress tolerance as well by the enhanced ROS scavenging potential and ABA sensitivity as reported in a rice study by Zhou, et al. [115], which observed, respectively, decreased and increased germination rates in *STK-OE* and *STK-KO* lines as compared to the WT under ABA treatment (1–6 μM). The study suggested that the *STK* overexpression-mediated upregulation of stress/ABA-activated protein kinase10 (*OsSAPK10*) could be behind the severe ABA-mediated seed germination repression in the *STK-OE* lines. *AGL35*, yet another MADS box gene, reportedly affects germination rates in certain hybrids only by affecting the endosperm cellularization process. The hybrid seeds derived from *AGL35*-defective *A. thaliana* (♀) and normal *A. arenosa* (♂) exhibit a much reduced germination rate while those derived from *AGL35*-defective *A. thaliana* (♀) and normal *A. lyrata* (♂) show a much higher rate as compared to the respective hybrid seeds derived from normal *A. thaliana* (♀) [116].

## 6. Gene-to-Factor Associations

To have a general overview on some of the major factors affecting plant growth and development, we chose hormones and biotic/abiotic factors to extract associated MADS box members in respective gene pools from the local reference database.

### 6.1. MADS Member–Hormone Association

We generated five hormone-associated gene pools, each on auxin, cytokinin, ethylene, gibberellin, and abscisic acid (Figure 6). Due to the low abundance of hormone-associated MADS studies, few of the gene pools showed MADS members with above-threshold hits. Nevertheless, the genes with as low as two hits, in most cases, appear to have functions true to the associated gene pool.

To mention a few of such examples, an auxin-associated gene pool member, *AGL62*, is known to induce auxin in the syncytial endosperm of a newly fertilized ovule (seed), a defect of which brings impaired auxin transport from the developing endosperm to integuments, leading to seed abortion [117]. *XAL2*/*AGL14* reportedly plays a role in auxin transport during Arabidopsis root development by upregulating *PIN1* and *PIN4* expression. Furthermore, its own expression is positively regulated by the auxin level in a positive feedback loop [57].

*SVP* showed at least one hit in all gene pools except the cytokinin-associated one (Figure 6b). The ethylene-associated gene pool had its single hit. Nevertheless, an SVP-focused study reported that its clade members show discrepancy in ethylene response-related ERE elements in their promoter with the SVP3 members (absent in Brassicaceae) harboring the highest number of ERE elements, suggesting its ethylene-dependent regulation [118]. The association of the *SVP* member with auxin has been briefly discussed earlier in the Section 5.9. Regarding its association with other hormones, we can take an apple study as an example, which showed that its *SVP* homologs, often referred to as *DORMANCY ASSOCIATED MADS-BOX* (*DAM*), exhibit the highest expression—brought upon mostly by the higher level of H3K4me3—during autumn. Their expression is positively affected by the ABA level in a positive feedback loop [119]. Furthermore, the study observed a significant overlap between the *SVP*/*DAM* target genes and the genes with differential H3K4me3 levels among the simulated-season-derived samples. The overlapped members included auxin and gibberellin (GA) biosynthesis as well as cell cycle and cell wall expansion-associated genes, among others, indicating a role of *SVP*/DAM in regulating the H3K4me3 level itself in a positive feedback loop. The study concluded that the elevated levels of auxin and GA as well as increased cell cycle progression are key to bud breaking during spring [119]. Notably, our analysis shows *SVP* hits above the threshold in the gene pools associated with GA and ABA, and a threshold-level hit in the auxin-associated one (Figure 6).

Ethylene associations with the MADS members were the lowest among all gene pools. *STK*, which returned a single hit, is often associated with the seed development and is an unusual gene to have association with ethylene. However, a tomato study with a modulated expression of its homolog *Sl-AGL11* showed that apart from obvious differences in the floral and fruit morphologies, the timing of the ethylene peak and ethylene level during the peak were widely different between the WT and *Sl-AGL11* overexpressing lines, which were correlated with the significant difference in the expression of the ripening associated genes [120].

### 6.2. MADS Members—Biotic/Abiotic Factor Association

Local reference database-derived independent gene pools were developed for biotic and abiotic factors, each associated with nutrients, defense (tolerance/resistance/susceptibility), light (response), salt/salinity, and osmosis (response). While the latter four did not return any MADS box members above the threshold, a few were returned for the former two (Figure 7). Interestingly, all the hits at and above the threshold in the nutrient gene pool were associated with the ANR1 clade except *SOC1*. As mentioned in an earlier section, *ANR1* and homologs play a role in nitrogen foraging. *SOC1* on the other hand reportedly responds to the changes in phosphorus and Sulfur [43]. *STK*, one of the members with the lowest hit in the pool, is often associated with ovule development and seed coat formation, and is one of the unlikely occurrences. However, a study associated with cell wall invertase (CWIN) reported that *STK* and other genes involved in ovule development are dependent on sugar signaling cues potentially received by the RLK members at the intracellular space [121]. The study proposed that CWIN may play a role in hydrolyzing the sucrose molecules at the intracellular spaces into glucose and fructose, which in turn may be sensed by the membrane-bound RLKs to regulate downstream genes involved in ovule development.

Among the genes associated with the defense, *SVP* returned with the highest hit (Figure 7b). The gene is known to play a role in age-related resistance (ARR) in Arabidopsis [122]. However, its role in biotic/abiotic stress has not been explored much. Nevertheless, a study related to the ACCase inhibitor herbicide (clodinafop-propargyl) tolerance by *Polygon fugax*, a weedy plant belonging to the Poaceae family, showed that the plant reportedly exhibits a positive correlation of its herbicide tolerance to *PfMADS11* expression and precocious flowering, even though the molecular mechanism behind the process remains yet to be elucidated [123]. The overexpression of the *SOC1*-like gene, *VcSOC1K*, in blueberry reportedly confers high pH tolerance to the plant [124]. Regarding *AP1*, a study on shade-tolerant orchid species *Cymbidium sinense* reported an expansion of AP1, SOC1, and SVP members [125]. However, whether such a case has any direct association with the shade tolerance remains unexplored. Regarding the light-associated MADS box members, the single gene *SOC1* was returned at a threshold-level hit. The gene is well known for its photoperiod response and expression fluctuations with the circadian rhythm. As reported in a poplar study, it plays an active role in seasonal ecodormant bud breaking as well. Furthermore, the study showed that plants overexpressing its *SOC1* homolog, *MADS12*, significantly induce much precocious budbreaking at long-day conditions without pre-chilling treatment via the downregulation of *GA2ox4*, a gene actively involved in GA degradation, during the process [32].

Our analysis returned *SOC1* hits in the heat-associated gene pool as well, albeit below the threshold. Its temperature responsiveness is often not highlighted. However, studies show that its photoperiodic response is further enhanced at a warmer temperature in plants [32]. Interestingly, *SOC1* showed hits to the salt/salinity-associated gene pool as well, although below the threshold. As reported in a study, stress-dependent dual-localizing *OXS2*, a zinc finger transcription factor essential for salt tolerance [126], plays an active role in activating *SOC1* by directly binding to its promoter during a stress condition in Arabidopsis. In a normal state, however, OXS2 is localized at the cytoplasm and promotes vegetative growth [127]. *SOC1* additionally showed a hit for the osmotic response-associated gene pool. The associated study carried out a functional characterization of *Ginkgo biloba*-derived TT16/GGM13 clade member *GbMADS9*, which showed that the plants overexpressing the gene exhibit better growth under high osmotic stress (as compared to WT) and lead to precocious flowering due to the increased expression of florigenic genes *FT*, *AP1*, *LFY*, and *SOC1* [70]. However, a relatively recent study suggests that *SOC1* itself may not have a direct effect on the process [128]. The involvement of *AGL21* in the regulation of osmotic stress is well studied. One such example includes an Arabidopsis study by Yu, et al. [108], which reported the hypersensitivity of the *AGL21* overexpressing lines to osmotic, ABA, and salt stresses during seed germination.

## 7. Trait-to-Factor Associations Bridged by MADS

While working with a specific phenotype, a general overview of potential genes linked with the factors associated with the phenotype would offer information on genetic layers and potential directionality of the genes’ action. Direct literature-derived information would be very helpful in such a case. Being one of the heavily studied gene groups in association with flowering, MADS box members are expected to have relatively richer information regarding their role in bridging the biotic and abiotic factor-derived cues to the process.

Flowering is a complex process. However, studies have often demonstrated that the ectopic expression of florigenic terminal genes is sufficient for floral induction in many cases, which often renders the transgenic plant phenotypically different/deformed as compared to its wild-type counterpart, indicating a potential genetic bottleneck behind the phenomenon. Such an effect is more pronounced in perennials [129,130,131,132]. Plants respond to the biotic and abiotic cues to allocate their resources according to their physiological need. When those processes are cut short or abruptly disturbed via a transgenic approach, such cues are less likely to be aligned in the plant, which could be the main reason behind such an aberrant phenotype.

Being a terminal developmental process in a plant’s life cycle, flowering commences either when the plant is fully mature or if there is risk-to-perish prior to its maturity due to unavoidable biotic/abiotic factors [133,134]. In other cases, the flowering frequency and intensity may decrease when there is ample fruit/seed set to secure the next generation through a negative feedback loop, which we discussed earlier in the Section 5.8 and Section 5.9. We screened MADS box members with such potentials of bridging external/internal cues to the flowering process. In total, there are eight separate gene pools—each with a potential role in bridging the flowering process to fruit/seed development—root development/biomass, nutrients, stress response, hormonal cues, seasonal changes, aging, and plant life cycle (Figure 8, outer gene pools). Even though five out of them returned genes above the threshold, all of them apparently show literature evidence.

We additionally checked potential multifactor integrator MADS box members (Figure 8, central gene pool) based on their frequency of occurrences in the aforementioned independent gene pools. *SOC1* and *FLC* showed the highest hit (seven each) followed by *AP1* (six) and *FUL* and *XAL1* (five each), roughly suggesting that their ectopic expression modulation may bring phenotypic abnormalities in transgenic plants. Such an assumption is partly corroborated by a transgenic study with *MtSOC1a* in Medicago (perennial plant), in which the overexpression lines not only exhibited a precocious flowering phenotype but showed increased shoot growth as well [135]. In a different study on soybean (annual plant), however, maize-derived *ZmSOC1* conferred shorter plant height with frequent abnormal flower development, but increased branching and pod numbers per plant among the overexpression lines as compared to the wild-type [136]. As mentioned earlier, its constitutive expression in Arabidopsis reportedly causes failure of seed development. It should be noted that *SOC1* is one of the key flowering pathway integrators [84]. *FLC* along with the majority of its clade members plays a role in the temperature/vernalization-dependent flowering process. *AP1* and its clade members (including *FUL*) function terminally in the flowering pathway, and *XAL1* mainly contributes to root growth and development as well as in the flowering process. While its defect brings significant delay in the process, its overexpression effect on flowering is not as significant likely because *XAL1* itself may not be sufficient to activate its target genes involved in the process [58,137].

Even though the majority of the MADS box members play a direct crucial role in floral development and some in vegetative-to-reproductive phase transition, they are not the only major players behind flowering-associated physiological processes. While the modulated expression of florigenic MADS box members often triggers plants to produce a new sink (flower), its state and further developmental progression would still necessitate proper alignment of the underlying physiological processes in the plant system. Similar comprehensive assessment particularly focusing on flowering rather than a particular gene group may offer relatively robust findings, returning with additional key players involved in the physiological processes during flowering.

## 8. Optimization Considerations for the Approach

During the literature data extraction and analysis, we customized our approach to better fit its result with the study findings. Below are some of those key customization parameters considered—

*Threshold calibration:* Thresholds for each analysis may depend on the volumes of the studies in the local reference database. Larger volumes of references along with higher threshold hit assignment may enhance reliability of the assessment. From our analysis, a threshold hit of at least three is sufficient to return a workable result from a representative local reference database.

*Choice of keywords/terms:* As observed in MADS box member assessment, several gene IDs may match with their respective clade IDs (e.g., *FLC*, *AP1*, *SOC1*). Hence, such IDs often return with higher hits. In such a case, their respective association with a particular trait could equally be trait-to-clade association in addition to trait-to-gene association. Furthermore, use of dual-meaning terms (e.g., light) may include higher false-positive hits. Use of exclusion for the search-term-associated unwanted phrases could circumvent the case. In rare cases, search terms may match with the unintended annotations used in the studies. One such example includes the occurrence of “AG” in naming an allele in a rice study [138], which was picked up in the gene pools associated with the gene *AG*. Use of a suitable (higher) threshold level would help reduce such unwanted ‘noise’ data.

*Analysis skewedness:* The pleiotropic gene pooling approach used in the current study basically depends on the independent trait-based gene pools used for the analysis and tends to have skewedness towards the most studied members as the prediction circles back to the holistic assessment of those independent gene pools derived from the same local reference database. Expanding reference database size may certainly help with circumventing such a case to some level. However, allowing some buffer zone (gray area) at both sides of the threshold and the manual inspection of the genes within the area are expected to enhance the analysis strength.

## 9. Significance and Application of the Approach

Research studies are often carried out in narrower niches of fields with more specific objectives as knowledge and technology advance over time. While it is beneficial to have a narrow study focus, it may sometimes leave obvious blind spots that would otherwise have been noticed. In other cases, not all studies are equally legible to all researchers. Additionally, while we gain expertise through knowledge and experience, subject matter experts with expertise may not always be available or accessible. The approach devised in the current study aims to circumvent such cases.

By using relevant keywords and constraints along with suitable threshold assignments, the current approach offers an alternative that provides an expert-like perspective on the subject matter of interest. Furthermore, it would offer an opportunity to gain a quick overview of the subject matter from multiple perspectives, which is often deemed crucial to the experts and non-experts alike during the initial phases of research and experimental design. The current approach is also useful for providing a data-based overview of any potential study biases, as observed between type I and type II MADS box members in this study.

## 10. Conclusions

Our assessment showed a clear disparity between studies associated with type I and type II MADS box members. While most of the MADS box-associated studies are flower- and fruit-focused, and MADS box members indeed have played a significant role in the evolution of angiosperms, our study suggests that there are additional avenues for their functional relevance in plants. We devised and used an approach to extract gene associations with various factors and developmental stages from the manually curated, MADS-focused local reference database (all the retrieved gene pool-associated data are provided in Appendix A). Such an approach is equally applicable to any other study of interest, whether it is focused on a particular gene, a specific trait, or any other topic of interest (for non-biological disciplines).

## Figures and Tables

**Figure 1 ijms-25-08233-f001:**
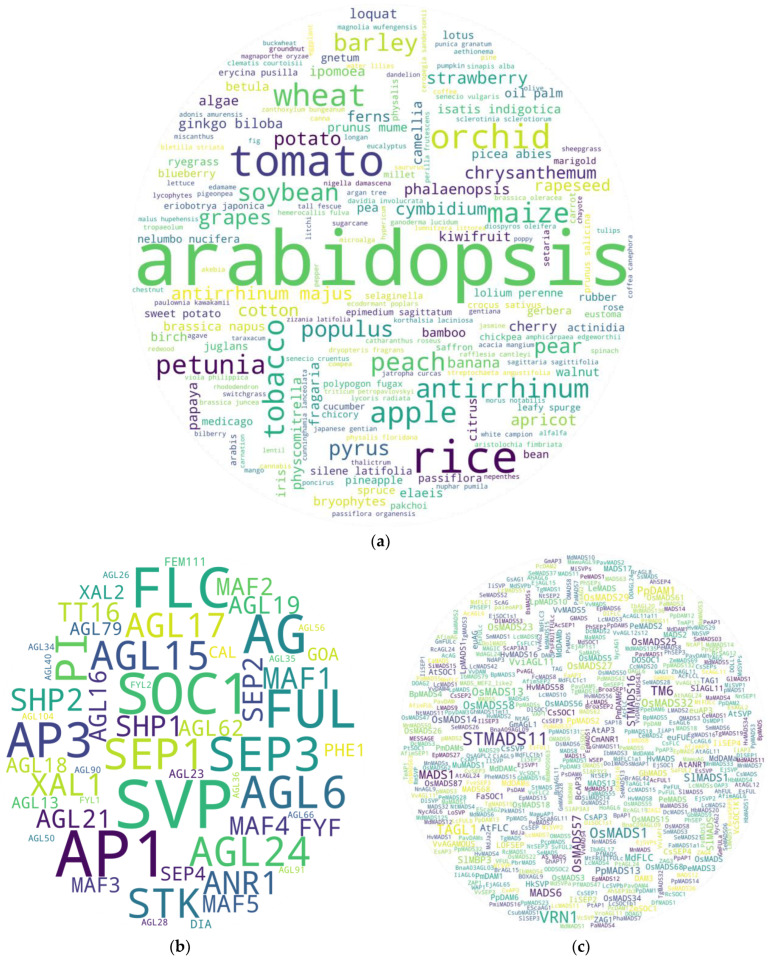
General overview of MADS box studies. (**a**) Study-to-organism associations ranging from 1 (several species) to 363 (Arabidopsis). (**b**) MADS box gene-ID-to-study association ranging from 1 (several genes) to 76 (*SVP*), (**c**) putative cross-species MADS box gene-ID-to-study association ranging from 1 (several genes) to 12 (*STMADS11*). Gene-word sizes are relative to their frequencies in each gene pool. {*Note:* High-resolution images of each subfigure have been provided with subfigure IDs in Appendix A}.

**Figure 2 ijms-25-08233-f002:**
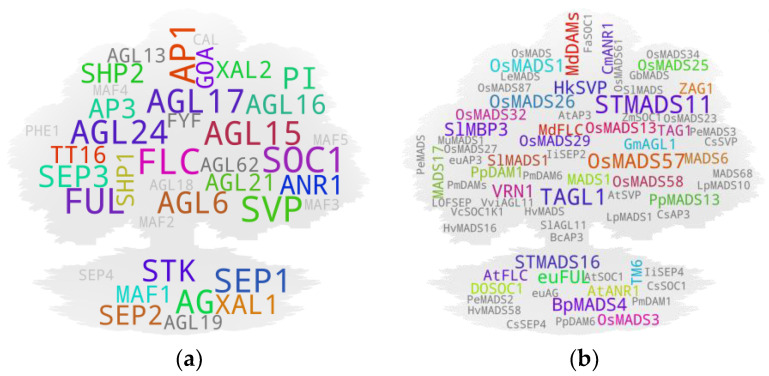
Genes with multi-organ associations. (**a**) Arabidopsis gene IDs associated with at least two of seven plant organs (root, shoot, leaf, apical meristem/SAM, flower, fruit, and seed). The genes shown had at least three hits for each of their respective organs. IDs with two hits among organs are in gray, and those with only one hit are in light gray. (**b**) Cross-species gene IDs associated with at least two of seven plant organs. The genes shown had at least two hits for each of their respective organs. IDs with only one hit are shown in gray. {*Note:* High-resolution images of each subfigure have been provided with subfigure IDs in Appendix A}.

**Figure 3 ijms-25-08233-f003:**
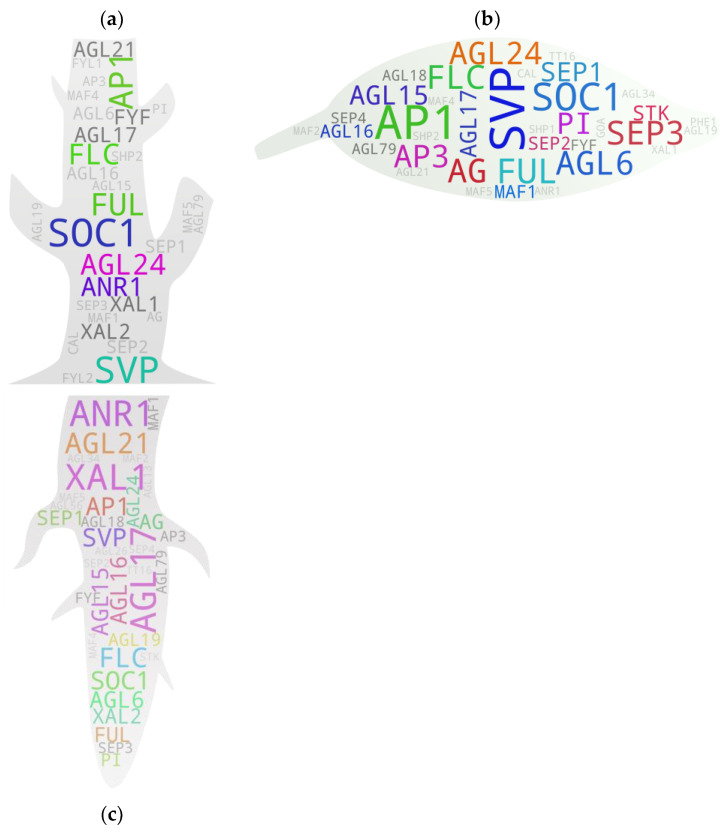
MADS box members associated with shoots (**a**), leaves (**b**), and roots (**c**). IDs with 1–2 hits are in light gray, and those with 3 hits are in dark gray. IDs with >3 hits are in any other random color. The text sizes are relative to their respective hit frequencies. {*Note:* High-resolution images of each subfigure have been provided with subfigure IDs in Appendix A}.

**Figure 4 ijms-25-08233-f004:**
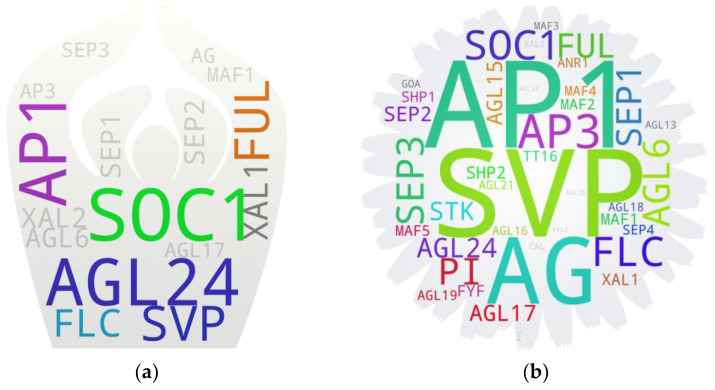
MADS box members associated with SAM (**a**), flower (**b**), ovule (**c**), and pollen (**d**). IDs with 1–2 hits are in light gray and those with 3 hits are in dark gray. IDs with >3 hits are in any other random color. Text sizes are relative to their respective hit frequencies. {*Note:* High-resolution images of each subfigure have been provided with subfigure IDs in Appendix A}.

**Figure 5 ijms-25-08233-f005:**
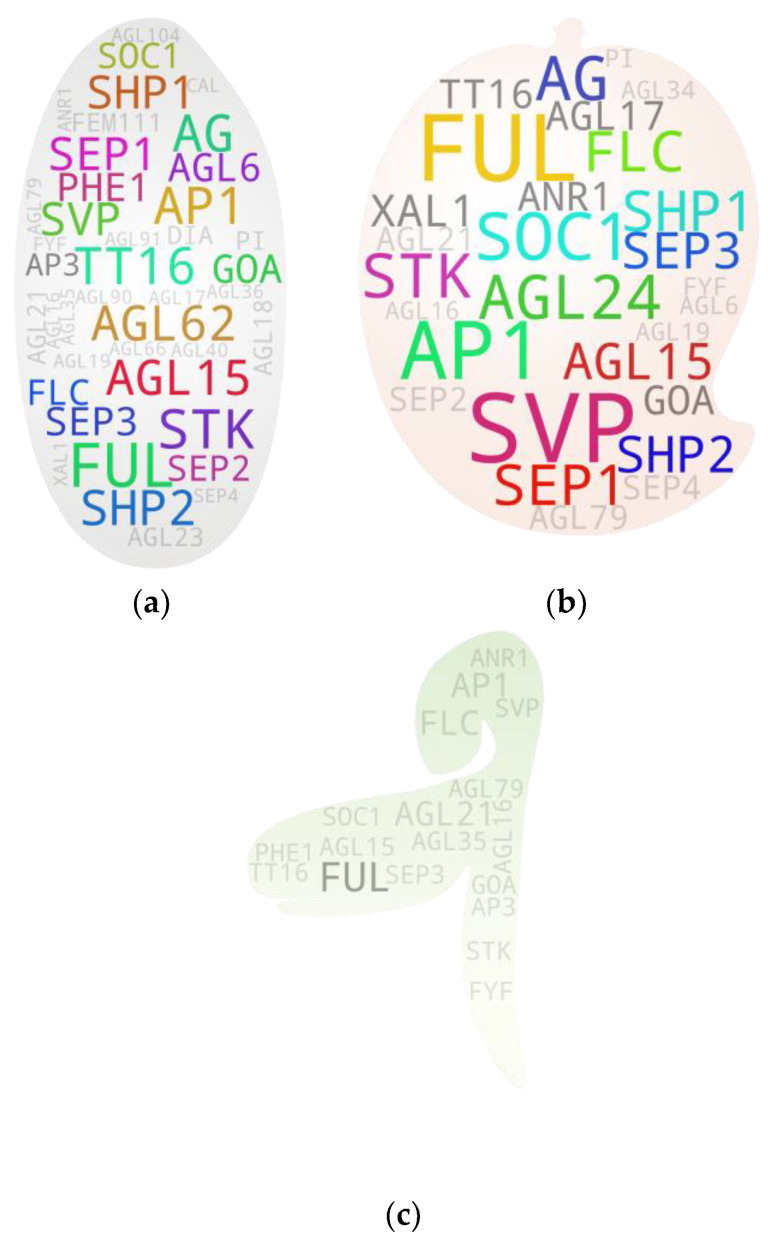
MADS box members associated with seeds (**a**), fruits (**b**), and seed germination (**c**). The IDs with 1–2 hits are in light gray and those with 3 hits are in dark gray. The IDs with >3 hits are in any other random color. The text sizes are relative to their respective hit frequencies. {*Note:* High-resolution images of each subfigure have been provided with subfigure IDs in Appendix A}.

**Figure 6 ijms-25-08233-f006:**
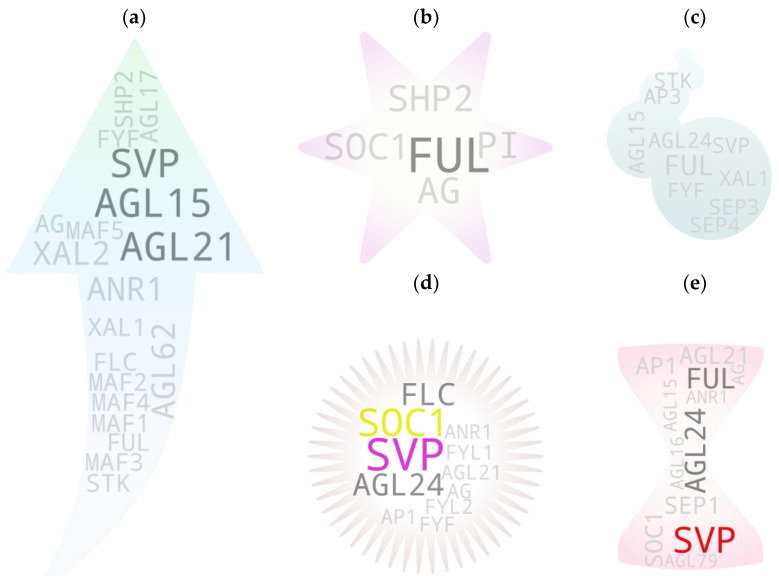
MADS box members associated with hormones. (**a**) Auxin, (**b**) cytokinin, (**c**) ethylene, (**d**) gibberellin, (**e**) abscisic acid. The IDs with 1–2 hits are in light gray and those with 3 hits are in dark gray. The IDs with >3 hits are in any other random color. The text sizes are relative to their respective hit frequencies. {*Note:* High-resolution images of each subfigure have been provided with subfigure IDs in Appendix A}.

**Figure 7 ijms-25-08233-f007:**
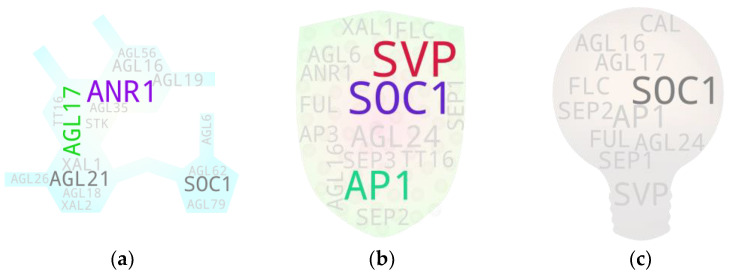
MADS box members associated with biotic and abiotic factors. (**a**) Nutrient response; (**b**) tolerance, resistance, or susceptibility response; (**c**) light response; (**d**) heat response; (**e**) salt response; (**f**) osmotic response. {*Note:* High-resolution images of each subfigure have been provided with subfigure IDs in Appendix A}.

**Figure 8 ijms-25-08233-f008:**
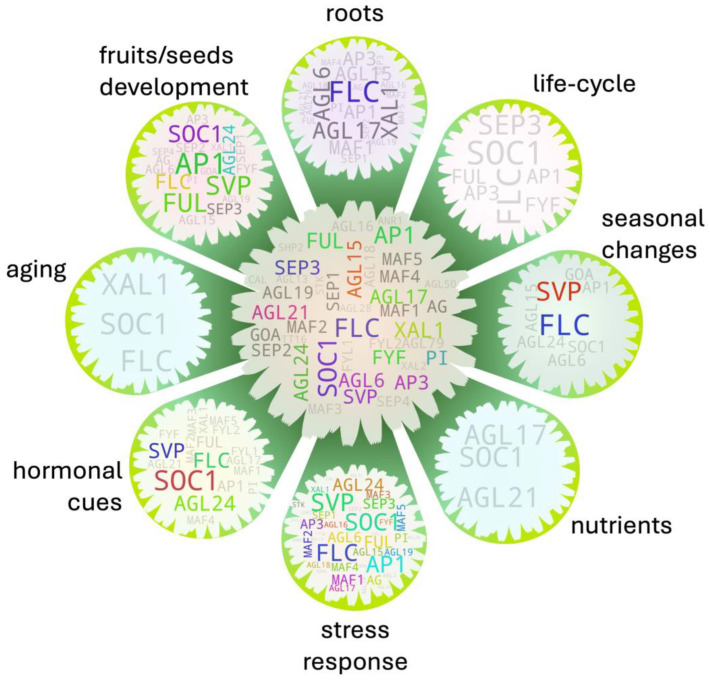
Trait-to-factor bridging MADS box members with the ‘flowering’ trait. Gene pools at the circles represent respective factor-to-flowering-associated MADS box members. The central gene pool was generated from all other gene pools to assess the most frequent MADS box members among them. The IDs with 1–2 hits are in light gray and those with 3 hits are in dark gray. The IDs with >3 hits are in any other random color. Their text sizes are relative to their respective hit frequencies within each gene pool. {*Note:* A high-resolution version of the image has been provided in Appendix A}.

## Data Availability

All the data used for and produced during the analysis have been included in the manuscript. The in-house script prepared during the analysis can be provided to researchers upon request.

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
