# Peer review of "An Overview on MADS Box Members in Plants: A Meta-Review"

_ijms, 2024, doi:10.3390/ijms25158233_

Round 1

Reviewer 1 Report

Comments and Suggestions for Authors

The MADS-box genes are very important in plant morphogenesis and have been the subject of an enormous amount of research. The main purpose of this paper, which is to organize this information, is very understandable, and I believe that the results are important.

1. First and foremost, the figures in this paper are very difficult to see and understand. It is necessary to clarify what is to be conveyed in each figure and redraw it.

2. Gene -to-major tissues growth association is very well summarized. However, previous studies of the MADS-box genes have revealed different mechanisms in monocotyledons and dicotyledons. If this is only for dicotyledons, this should be clearly stated. As a review, it would be better to organize and present the case of monocotyledons as well.

Author Response

The authors would like to thank the reviewer for allocating precious time and effort to review the submitted manuscript. The manuscript has now been revised for its written language and content. Two versions of the manuscript have been provided for the reviewers. One of them (supplied as Non-published Material)  contain all deleted (strikethrough red text) and added texts (blue underlined). The second one contains publication-ready contents (unless the reviewers find additional avenues for improvement) with all newly made changes highlighted in cyan. Reviewer’s comments have been addressed point-by-point below.

Reviewer1: The MADS-box genes are very important in plant morphogenesis and have been the subject of an enormous amount of research. The main purpose of this paper, which is to organize this information, is very understandable, and I believe that the results are important.

Authors’ Response: Authosr convey their gratitude for the reviewer’s understanding and commendation. The purpose of the article was also to establish the use case of a direct literature based strategy while preparing a review on the topic of interest, which was ‘MADS box genes’ in the submitted manuscript.

Reviewer1: First and foremost, the figures in this paper are very difficult to see and understand. It is necessary to clarify what is to be conveyed in each figure and redraw it.

Authors’ Response: Authors would like to thank the reviewer for the query and apologize for any misunderstandings regarding the presented figures. The IDs in most of the figures were intentionally made dim based on the set threshold as they were less likely to represent the associated gene pools. Additionally, the smaller gene ID texts represent their less repetitiveness among the literatures, suggestive of their less representativeness to the gene pool. To address the similar issue for the curious readers, we have provided the lists of each gene pool along with their frequency numbers in Supplementary Dataset 1.

Reviewer1: Gene -to-major tissues growth association is very well summarized. However, previous studies of the MADS-box genes have revealed different mechanisms in monocotyledons and dicotyledons. If this is only for dicotyledons, this should be clearly stated. As a review, it would be better to organize and present the case of monocotyledons as well.

Authors’ Response: Authors would like to thank the reviewer for the query. Authors agree with the reviewer’s argument that certain MADS box members have specialized their functions in certain lineages of plants like dicots, monocots (or angiosperms as a whole), ferns, etc. In currently presented literature-based review manuscript however, the authors did not particularly set any preference to certain plant lineages. Hence, the plants with lesser MADS box associated studies are expectedly discussed at lesser frequency. Nevertheless, the authors would like to point that monocot references have been frequently discussed throughout the manuscript (kindly check for the discussions related to rice, maize, wheat, etc.). To avoid similar confusion among the readers, we have mentioned about what to expect from this review in the very first paragraph (Kindly check Page 1, lines 27-29: “…. The contents of this article….. in article title.”).

Reviewer 2 Report

Comments and Suggestions for Authors

This paper summarises a meta-study approach to extract and review literature related to MADS-box gene family in plants in a relatively unbiased way. 

The abstract summarises the paper well, including some background, the approach, and the major findings. The language would benefit from further editing. I have added some suggested edits below. The introduction starts with some general overview of the approach and the study system, including justification for a function-led review of the MADs box gene family. The methods are mostly complete but some further clarifications are required as specified below. The multiple tissues criterion for identifying pleiotropic genes seems reasonable. Is there evidence for this approach in the literature for further justification? Each tissue or other type of association or combination of associations is then discussed in turn with nice development of examples from the supporting literature. The reader is referred to existing reviews, such as the role of MADS box genes in floral induction, to avoid redundant discussion but examples from these papers where they deal with primary literature should perhaps be briefly described alongside other examples (e.g. P417-421). The review does well to identify understudied topics deserving of more future attention, thereby serving a valuable purpose. This point is considered again in the overall conclusions and poses a useful consideration of wider interest beyond this specific review topic. 

Specific comments

L2 The title is a little cryptic. What is meant by "functional" here?

L10 Rewrite "box members their cross-trait" as "box members' cross-trait"

L16 Rewrite "discusses on the roles" with "discusses the roles"

L19 Drop "etc." here and elsewhere. 

L21-22 Rewrite "is expected applicable" as " is expected to be applicable"

L24 There is scope for more keywords.

L33-34 This opening statement could be expanded with a little more context about each of the four "founder proteins. 

L49 Should "interplay of known members to the hormonal cues" be "interplay of known members with hormonal cues"?

L66 Not sure what is meant by "proceeded for the gene-to-study association analyses"

L67 List the constraints used here.

L80 Typo rewrite "amont" as "amongst"

Figure S1 Clarify in the legend if the sequences in this phylogeny are all from Arabidopsis or a mix of species. 

L129-144 Break up long sentences here for readability.

L184-187 This statement seems focused on flowering. Why is it discussed in the shoot section?

L189 Figure 1a. Some frequent shoot associated genes such as SOC1, AP1 do not seem to be discussed in the relevent text section. Why is this?

L267-271 Split up this long sentence.

L329 Use "pollen" as plural term. 

L391-393 Develop these examples here.

L408 What is meant by "gene pool" in this context?

L417-421 It would be better to work these studies into the previous discussion with examples of specific observations from these papers.

L499-500 Unclear what is meant by "harboring the most suggesting its ethylene-dependent regulation"

L540-542 Missing panel f label.

L607-608 I would drop "majority of the genes... associated gene pools" as all presented genes have literature evidence supporting their inclusion as threshold simply refers to frequency of citation.

L612-613 It is not clear why transgenic plants for genes responsible to multiple cues would have "less phenotypic abnormalities" and not more.

Supplementary files. Multiple versions of some tables and databases appear to have been uploaded. Check which files are the final versions to accompany this paper. 

Comments on the Quality of English Language

I recommend careful language of this paper as part of revisions.

Author Response

Response to the Reviewer 2

The authors would like to thank the reviewer for allocating precious time and effort to review the submitted manuscript. The comments and suggestions offered by the reviewer were pivotal on addressing the shortcomings and improving the manuscript further. The manuscript has now been revised for its written English as well as contents as suggested by the reviewer. Two versions of the manuscript have been provided for the reviewers. One of them (supplied as Non-published Material) contain all deleted (strikethrough red text) and added texts (blue underlined). The second one contains publication-ready contents (unless the reviewers find additional avenues for improvement) with all newly made changes highlighted in cyan. Reviewer’s comments have been addressed point-by-point below.

Reviewer2: This paper summarises a meta-study approach to extract and review literature related to MADS-box gene family in plants in a relatively unbiased way.

Authors’ Response: Authors would like to convey their gratitude to the reviewer for the commendation. Part of the study objective was to present relatively unbiased description which the reviewer has correctly perceived.

Reviewer2: The abstract summarises the paper well, including some background, the approach, and the major findings. The language would benefit from further editing. I have added some suggested edits below. The introduction starts with some general overview of the approach and the study system, including justification for a function-led review of the MADs box gene family. The methods are mostly complete but some further clarifications are required as specified below.

 Authors’ Response: Authors would like to convey their gratitude to the reviewer for the commendation and detailed feedback from the reviewer. The resubmitted manuscript has now been thoroughly edited for its written English during which the reviewer’s additional suggestions have also been addressed.

Reviewer2: The multiple tissues criterion for identifying pleiotropic genes seems reasonable. Is there evidence for this approach in the literature for further justification?

 Authors’ Response: Authors would like to thank the reviewer for the comment. The authors attempted to skim the information from the published literatures based on the basic definition of pleiotropic gene which the reviewer have perceived well. The authors are not aware of similar approach on pleiotropic gene prediction/sorting solely based on the publications. Hence, part of the manuscript’s objective was also to establish the literature-based assessment to skim information which would not otherwise have been noticed via specific gene/plant/function-targeted studies.

Reviewer2: Each tissue or other type of association or combination of associations is then discussed in turn with nice development of examples from the supporting literature. The reader is referred to existing reviews, such as the role of MADS box genes in floral induction, to avoid redundant discussion but examples from these papers where they deal with primary literature should perhaps be briefly described alongside other examples (e.g. P417-421).

Authors’ Response: Authors would like to thank the reviewer for the comment. As the reviewer has well perceived, the authors have mentioedn specific study/studies on the discussion topic so as not to deviate too far from the discussion thread while offering the interested readers a quick look-up on the topic-specific study/review. While doing so, the authors have often attempted to maintain minimal description. After reviewer’s suggestion however, the revised manuscript now includes additional arguments at relevant places [Page 7, lines 250-254; Page 12, lines 401-409; Page 12-13, lines 434-443].

Reviewer2: The review does well to identify understudied topics deserving of more future attention, thereby serving a valuable purpose. This point is considered again in the overall conclusions and poses a useful consideration of wider interest beyond this specific review topic.

Authors’ Response: Authors would like to convey gratitude for the reviewer’s comendation and also for nicely summarizing the study strategy, aim, and scope in a concise yet best way possible.

Specific comments

Reviewer2: L2 The title is a little cryptic. What is meant by "functional" here?

Authors’ Response: The authors thank the reviewer for pointing it out. The title now reads “Overview on MADS box members in plants: A meta-review” [Page 1, line 2]

Reviewer2: L10 Rewrite "box members their cross-trait" as "box members' cross-trait"

Authors’ Response: The authors thank the reviewer for pointing it out. Suggested change has been adapted [Page 1, line 10].

Reviewer2: L16 Rewrite "discusses on the roles" with "discusses the roles"

Authors’ Response: The authors thank the reviewer for pointing it out. Suggested change has been adapted [Page 1, line 16]

Reviewer2: L19 Drop "etc." here and elsewhere. 

Authors’ Response: The authors thank the reviewer for pointing it out. Suggested change has been made [Page 1, line 18, 19; Page 2, line 62, 76, 83, 86, 91; Page 4, line 119, 147; Page 5, line 168; Page 7, line 248; Page 10, line 376; Page 19, line 675]

Reviewer2: L21-22 Rewrite "is expected applicable" as " is expected to be applicable"

Authors’ Response: The authors thank the reviewer for pointing it out. Suggested change has been adapted [Page 1, line 21].

Reviewer2: L24 There is scope for more keywords.

Authors’ Response: The authors thank the reviewer for pointing it out. Additional keyword ‘biotic and abiotic factors’ has now been included [Page 1, line 23].

Reviewer2: L33-34 This opening statement could be expanded with a little more context about each of the four "founder proteins”

Authors’ Response: The authors thank the reviewer for the suggestion. The sentence has been modified to incorporate relevant information/basis behind ‘MADS’ annotation [Page 1, line 32-35].

Changes made: “…. MADS represents the four fabulous founder homeotic proteins: MCM1 (from Saccharomyces cerevisiae), AGAMOUS (from Arabidopsis thaliana), DEFICIENS (from Antirrhinum majus), and SRF (from Homo sapiens), which was first observed and annotated as such by Schwarz-Sommer, et al. [1] ….”

Reviewer2: L49 Should "interplay of known members to the hormonal cues" be "interplay of known members with hormonal cues"?

Authors’ Response: The authors thank the reviewer for the suggestion. Suggested change has been made [Page 2, line 49].

Changes made: “…. the interplay of known members with hormonal cues ….”

Reviewer2: L66 Not sure what is meant by "proceeded for the gene-to-study association analyses"

Authors’ Response: The authors thank the reviewer for pointing it out. The sentence has now been modified to avoid such ambiguity [Page 2, line 66].

Changes made: “…. and used for the gene-to-study association analyses ….”

Reviewer2: L67 List the constraints used here.

Authors’ Response: The authors thank the reviewer for the suggestion. The constraints used to screen the literatures for specific gene pools have now been included as Supplementary table S1. The main text now contains its reference [Page 2, line 70].

Reviewer2: L80 Typo rewrite "amont" as "amongst"

Authors’ Response: The authors thank the reviewer for pointing it out. Suggested change has been adapted [Page 2, line 81].

Reviewer2: Figure S1 Clarify in the legend if the sequences in this phylogeny are all from Arabidopsis or

Authors’ Response: The authors thank the reviewer for pointing it out. Suggested change has now been adapted [Page 20, line 721].

Changes made: “Figure S1: Arabidopsis-derived MADS box member TCOFFEE ….”

Reviewer2: L129-144 Break up long sentences here for readability.

Authors’ Response: The authors thank the reviewer for pointing it out. Suggested change has now been adapted [Page 4, lines 130-147].

Changes made: “Such examples include its involvement in meristem determinacy by negatively regulating AP2 in the developing inflorescence [17], and in apical hook opening modulation by negatively regulating the expression of growth-promoting genes in Arabidopsis [18]. Additionally, the involvement of its homolog from birch [19] and several other plants in precocious flowering has been observed. Yet additional instances include the role of rice FUL homolog in normal seed development by regulating at least two key genes involved in starch synthesis, OsAGPL2 and WAXY [18], and the crucial role of its tomato orthologs FUL1 and FUL2 in tomato fruit ripening, potentially by forming a tetramer complex with additional MADS box members RIN and AGL1 [20]. The tomato FUL1/2, along with an additional MADS member MBP20 (a SEP-like gene), has also been attributed with roles in vegetative-to-reproductive transition and inflorescence architecture regulation [21]. In rice as well, AP1/FUL homologs (OsMADS14, OsMADS15, and OsMADS18) and a SEP homolog (PAP2) reportedly confer floral meristem identity [22]. Another study further reported the ABA-responsiveness of OsMADS18 and its involvement in various developmental features including germination, tillering, and inflorescence architecture [23]. In soybeans, a study reported the involvement of a FUL homolog haplotype, GmFULa, in plant biomass and seed yield without affecting flowering time [24].”

Reviewer2: L184-187 This statement seems focused on flowering. Why is it discussed in the shoot section?

Authors’ Response: The authors thank the reviewer for the comment. The statements in question were to convey a message that FLC like genes may also confer effect comparable to that of FLC. However, as the reviewer pointed, it apparently was ambiguous. The statements have now been omitted (Page 5; line 186).

Reviewer2: L189 Figure 1a. Some frequent shoot associated genes such as SOC1, AP1 do not seem to be discussed in the relevent text section. Why is this?

Authors’ Response: The authors thank the reviewer for the comment. Authors aimed to check the relevance of genes with the near threshold values. However, after the reviewer’s suggestion, relevant arguments have been added so as not to leave the section incomplete [Page 5, lines 186-191].

Changes made: “…… Regarding SOC1, positive role of its ortholog, MADS12, in accelerated shoot growth was reported in poplar by its repressive effect on GA2ox4, a negative regulator of shoot growth [32]. AP1 and homologs, on the other hand, act in opposite spectrum as reported for AP1/FUL ortholog PgMADS1 in Panax ginseng [33]. ANR1, a MADS member involved in nitrate regulation and signaling in root, confers positive effect in shoot growth as reported in an Arabidopsis study [34].”

Reviewer2: L267-271 Split up this long sentence.

Authors’ Response: The authors thank the reviewer for pointing it out. Suggested change has been adapted (Page 7; lines 275-279).

Changes made: “…… FUL, on the other hand, has been attributed with a role in global proliferation arrest of active meristems by directly repressing members of AP2 clade, the ERF members. This negatively regulates the flowering and flower development process, which would otherwise repress the repressors of WUSCHEL, a key gene involved in meristem maintenance [62].”

Reviewer2: L329 Use "pollen" as plural term. 

Authors’ Response: The authors thank the reviewer for pointing it out. Suggested change has been adapted (Page 9; lines 339).

Reviewer2: L391-393 Develop these examples here.

Authors’ Response: The authors thank the reviewer for pointing it out. Suggested change has been adapted (Page 12; lines 401-409).

Changes made: “There are several studies on seeds associated MADS members. To name few of them here, Bemer, et al. [93] carried out an extensive assessment of type I MADS box members ex-pression pattern in ovule and developing seeds. Some of the seed-expressed MADS-box members reported in the study include PHE1/2 (early developing endosperm), AGL28 (developing embryo), AGL46 (developing endosperm), AGL35 (chalazal endosperm) among others. Ehlers, et al. [94] reported the roles of SHP1 and SHP2 in endosperm formation and seed coat development in developing seeds. Coen, et al. [95] observed that TT16 and STK act as master regulator of sub-epidermal integument cell layer patterning in developing seeds.”

Reviewer2: L408 What is meant by "gene pool" in this context?

Authors’ Response: The authors thank the reviewer for pointing the ambiguous statement. Changes have been made to circumvent that (Page 12; lines 424-425).

Changes made: “… potential involvement of flowering related MADS box members present in the fruit as-sociated gene pool in fruit dependent feedback loop.”

Reviewer2: L417-421 It would be better to work these studies into the previous discussion with examples of specific observations from these papers.

Authors’ Response: The authors thank the reviewer for the suggestions. Suggested changes have been adapted (Page 12-13; lines 438-443).

Changes made: “…. process in tomato and other fruit crops. The study by Busi, et al. [103] observed the expression of TAGL12, an XAL1 ortholog through all stages of fruit development. The study by Wang, et al. [104] suggested crucial roles of DlSTK, DlSEP1/2, and DlMADS53 (putative AGL62 and DIA ortholog) in the plant’s fruit growth and ripening. The study by Li, et al. [105] discusses on several genes which are grouped within the gene pool of current study.”

Reviewer2: L499-500 Unclear what is meant by "harboring the most suggesting its ethylene-dependent regulation"

Authors’ Response: The authors thank the reviewer for pointing out the ambiguous sentence. Changes have been made to circumvent it (Page 12-13; lines 521-522).

Changes made: “…. in their promoter with the SVP3-members (absent in Brassicaceae) harboring the highest number of ERE elements suggesting….”

Reviewer2: L540-542 Missing panel f label.

Authors’ Response: The authors thank the reviewer for pointing out. The labels have now been corrected and missing one has been included (Page 16; lines 564).

Changes made: “…. (c) light response, (d) heat response, (e) salt response, (f) osmotic response.”

Reviewer2: L607-608 I would drop "majority of the genes... associated gene pools" as all presented genes have literature evidence supporting their inclusion as threshold simply refers to frequency of citation.

Authors’ Response: The authors thank the reviewer for pointing out. The labels have now been corrected and missing one has been included (Page 17; lines 628-629).

Changes made: “…. Even though five out of them returned genes above threshold, all of them apparently show literature evidence.”

Reviewer2: L612-613 It is not clear why transgenic plants for genes responsible to multiple cues would have "less phenotypic abnormalities" and not more.

Authors’ Response: The authors thank the reviewer for pointing out the ambiguous sentence. The aim was to convey message that defect in these genes is likely to always bring some level of phenotypic abnormality even if it is at minimal level. Changes have been made to the sentence to circumvent such ambiguity (Page 17; lines 634).

Changes made: “…. roughly suggesting that their ectopic expression modulation may bring phenotypic abnormalities in the transgenic plants.”

Reviewer2: Supplementary files. Multiple versions of some tables and databases appear to have been uploaded. Check which files are the final versions to accompany this paper. 

Authors’ Response: The authors thank the reviewer for pointing the ambiguity. The supplemental data present in dataset 1 are of the gene pools described in the main text. Each sheet has now been added with title to avoid any ambiguity for the readers. Additional supplementary table containing constraint terms has now been included.

Round 2

Reviewer 1 Report

Comments and Suggestions for Authors

Reviewers still question the way the figures in this paper are represented. Will readers be able to understand the figures? For example, in Figure 1, some of the text is so small that it is almost impossible to decipher. What is the point of showing these? If the letters are not decipherable, they have no meaning even if they are in the figure. For example, how about including only the top 10 or top 20?

The authors responded as follows: The IDs in most of the figures were intentionally made dim based on the set threshold as they were less likely to represent the associated gene pools. The author must explain in the text so that the meaning can be easily understood in this sentence.

Related to the above explanation, for example in Figure 2b, the author needs to explain in clear detail what is the basis for this comparison of MADS box genes in different plant species.

Assuming that there was evidence that the 773 research papers could be compared, the authors need to explain in plain language what they were clarifying in this paper in the first place.

Author Response

The authors would like to convey gratitude to the reviewer for such a quick response back. The changes have been made to the manuscript and data based on the reviewer’s suggestions. The comments are addressed on point-by-point basis below.

Reviewer1: Reviewers still question the way the figures in this paper are represented. Will readers be able to understand the figures? For example, in Figure 1, some of the text is so small that it is almost impossible to decipher. What is the point of showing these? If the letters are not decipherable, they have no meaning even if they are in the figure. For example, how about including only the top 10 or top 20?

Authors’ Response: The authors would like to apologize for the oversight and thank the reviewer for adding arguments to make the point. The authors completely agree with the reviewer, as presenting illegible figures would not only make the curious readers perplexed but also make them question whether the authors really understood what they were doing. To circumvent such a case, the authors have now included all gene cloud figures and sub-figures (from figure 1 to figure 8) as part of supplementary materials in high resolution. While the reviewer’s suggestion of including only the top 10 or 20 could be a possibility, in some cases, several of the genes have the same hit numbers, making it difficult to pick them in an unbiased way. Hence, for the curious readers, like the reviewer himself, the now included supplementary images along with their respective hit data provided in the supplementary dataset 1 are expected to offer a clearer view. Additionally, all figure legends now include references to high resolution images. [Page 3, lines 109–110; Page 4, lines 126–127; Page 6, lines 207–210; Page 8, lines 301-303; Page 11, lines 294–396; Page 14, lines 526–527; Page 16, lines 581–582; Page 18; lines 673-674]

Reviewer1: The authors responded as follows: The IDs in most of the figures were intentionally made dim based on the set threshold as they were less likely to represent the associated gene pools. The author must explain in the text so that the meaning can be easily understood in this sentence.

Authors’ Response: The authors would like to thank the reviewer for the suggestion. Even though figure legends contained such an explanation, the main text apparently lacked such a clear description before jumping into the data description. Texts have been added as per the reviewer's suggestion [Page 2, lines 77–79].

Changes made: “…While visualizing the data, the IDs with hit numbers below the threshold were grayed out to make the IDs with above-threshold hits stand out..”

Reviewer1: Related to the above explanation, for example, in Figure 2b, the author needs to explain in clear detail what is the basis for this comparison of MADS box genes in different plant species.

Authors’ Response: The authors would like to thank the reviewer for pointing it out. While the basis of the generating figure was the same as that of other figures, not clarifying the method used to screen those cross-species related genes might have made it obscure to understand. The authors have now provided the keywords and search terms used to screen the gene IDS, plant organisms, tissues, and cross-species gene terms as supplementary dataset 2 (the dataset 2 of the earlier version has now been reannotated as dataset 3). Relevant changes were made in the main text as well [Page 2; lines 70–75].

Changes made: “…For general gene IDs, we used the Arabidopsis MADS box member gene IDs and their respective synonymous IDs as search keywords. For cross-species-specific MADS box gene screening, we used wildcards. The complete list of gene ID search keywords, wildcards, as well as organism and plant organ search terms, is provided in Supplementary Dataset 2....”

Reviewer1: Assuming that there was evidence that the 773 research papers could be compared, the authors need to explain in plain language what they were clarifying in this paper in the first place.

Authors’ Response: The authors would like to thank the reviewer for pointing it out. The authors would like to apologize for the oversight. The first paragraph of the main text now incorporates the suggested changes [Page 1; lines 29-31]. Additionally, the literatures belonging to the used local database have been provided in Supplementary Dataset 1.

Changes made: “…The study has attempted to establish a direct, literature-based approach to conducting a literature review using relevant search keywords, and constraint terms....”

Round 3

Reviewer 1 Report

Comments and Suggestions for Authors

Scientific papers are ultimately the responsibility of the author for publication, and it is the author's responsibility whether or not he or she incorporates the reviewers' opinions.